# Leveraging Biomaterial Platforms to Study Aging-Related Neural and Muscular Degeneration

**DOI:** 10.3390/biom14010069

**Published:** 2024-01-04

**Authors:** Veronica Hidalgo-Alvarez, Christopher M. Madl

**Affiliations:** Department of Materials Science and Engineering, University of Pennsylvania, Philadelphia, PA 19104, USA; hvero@seas.upenn.edu

**Keywords:** biomaterials, aging, 3D neural models, 3D muscular models

## Abstract

Aging is a complex multifactorial process that results in tissue function impairment across the whole organism. One of the common consequences of this process is the loss of muscle mass and the associated decline in muscle function, known as sarcopenia. Aging also presents with an increased risk of developing other pathological conditions such as neurodegeneration. Muscular and neuronal degeneration cause mobility issues and cognitive impairment, hence having a major impact on the quality of life of the older population. The development of novel therapies that can ameliorate the effects of aging is currently hindered by our limited knowledge of the underlying mechanisms and the use of models that fail to recapitulate the structure and composition of the cell microenvironment. The emergence of bioengineering techniques based on the use of biomimetic materials and biofabrication methods has opened the possibility of generating 3D models of muscular and nervous tissues that better mimic the native extracellular matrix. These platforms are particularly advantageous for drug testing and mechanistic studies. In this review, we discuss the developments made in the creation of 3D models of aging-related neuronal and muscular degeneration and we provide a perspective on the future directions for the field.

## 1. Introduction

Aging results in a progressive decline in tissue function and is accompanied by an increased risk of disease, posing high health risks and compromising quality of life [1,2]. A common consequence of aging is diminished muscle function, which can limit patient mobility and increase the risk of falls and subsequent injury [3,4]. Many common aging-associated disorders are neurodegenerative, leading not only to cognitive issues but also to mobility problems due to the interconnected nature of the nervous and muscular systems [5,6,7].

Neurodegenerative disorders are characterized by a progressive loss of neurons or neuronal function. The most common neurodegenerative diseases include Alzheimer’s and Parkinson’s diseases [8]. While age is the main risk factor, recent studies have revealed that both genetic and environmental factors are also significant contributors to the onset of these disorders [8,9]. Both Alzheimer’s and Parkinson’s are characterized by an aberrant accumulation of protein aggregates [10]. In the case of Alzheimer’s, the main features are the accumulation of amyloid β (Aβ) proteins in the form of plaques and the formation of neurofibrillary tangles (NFTs) composed of hyperphosphorylated tau [9]. The formation of plaques disrupts the hippocampal circuitry and therefore impairs the long-term consolidation of new memories [11]. As for Parkinson’s, the main histological manifestations of this disease include the formation of fibrillar aggregates of α-synuclein, known as Lewy bodies, and the loss of dopaminergic neurons in the substantia nigra [12] (Figure 1). Symptoms develop when 70–80% of dopaminergic neurons have been degraded, and include bradykinesia, resting tremors, and increased muscular rigidity [13,14].

Aging-related nerve cell degeneration can also result in the loss of muscle mass and function. This is the case when motor neurons are affected by degenerative processes, compromising the structural and functional integrity of the neuromuscular junction (NMJ) [15]. The NMJ is a cholinergic synapse of the peripheral nerve system that is established between an efferent motor neuron and a skeletal muscle fiber [16,17,18]. It is composed of three elements: the motor nerve terminal (presynaptic), the synaptic basal lamina, and the muscle fiber (postsynaptic) [19]. When an action potential reaches the motor nerve terminal, voltage-dependent calcium channels open and facilitate the entry of calcium into the neuron. This triggers the release of acetylcholine (ACh) into the synaptic cleft, where it diffuses through the basal lamina toward the muscle fiber. There, ACh binds its postsynaptic receptors (AChRs), starting a cascade of molecular events that results in the contraction of the muscle fiber [17,19,20]. During aging, this process is hindered by the degeneration of NMJs and the subsequent muscle denervation, as well as the failure of re-innervation mechanisms. This impaired NMJ function, together with the reduction in regenerative capacity that is characteristic of aged tissues, leads to the decline in muscle mass and function known as sarcopenia [15,19,21]. Furthermore, aging is a major risk factor for neuromuscular diseases that affect the NMJ, such as amyotrophic lateral sclerosis (ALS) [22]. This disorder leads to muscle weakness and atrophy, gradual paralysis, and respiratory failure [23].

Aging not only affects the structure and function of cells directly, but it also exerts an indirect effect by altering the extracellular matrix (ECM) that supports them [24]. Changes in the ECM structure, composition, and mechanical properties lead to alterations in cell signaling pathways that ultimately affect gene expression [25]. Neurodegenerative diseases are also associated with changes in the ECM [26]. Alzheimer’s disease (AD) has been reported to cause an increased production of hyaluronic acid (HA), chondroitin sulfate proteoglycans (CSPGs), and tenascin, and a reduction in the levels of reelin [27]. These alterations in the ECM composition lead to demyelination and protection of Aβ plaques from degradation [27]. In the case of Parkinson’s, a differential expression has been detected for collagens IV and VI, laminins, integrins, tenascin, annexins, brevican, neurocan, versican, and decorin [13,28]. While these changes have been detected in the brain, ECM alterations associated with neuromuscular aging or disease occur in the NMJ basal lamina, the specialized matrix present at the synaptic cleft of this synapsis. Studies conducted in mouse models have shown that the deletion of the proteoglycan agrin leads to a significant reduction in the levels of laminin α4 and α5 compared to control NMJs, and that this causes age-related alterations in the NMJ such as the fragmentation of postsynaptic AChRs or the denervation of endplates [29]. Changes in the distribution of these proteins with age have also been reported: mRNA expression analysis has shown a re-localization of agrin and laminin from synaptic regions to perisynaptic locations [29]. The substantial effect on NMJ aging of the alterations in agrin and laminin expression and location can be explained by their role in synaptic formation, organization, maturation, and maintenance [20]. Beyond the NMJ, the ECM of the skeletal muscle is known to change with aging, for instance, through fibrotic ECM deposition and crosslinking and through altered composition of the protein components of the ECM [21,30,31,32].

Given the important role that the ECM plays in the pathophysiology of neuromuscular disorders, it is essential to include this element in the in vitro models developed to study these diseases. Cell culture within 3D matrices is often required to maintain proper cell morphology, proliferation, mechanosensitive signaling, and epigenetic regulation [33,34]. Thus, traditional 2D cell culture models based on monolayer cell growth on rigid plastic surfaces typically fail to recapitulate the structure and function of human tissue [35,36,37]. On the other hand, bioengineered 3D models are emerging as a promising alternative to 2D cell culture due to their biomimetic structure that recapitulates the cell microenvironment [38,39]. This is achieved using biomaterials as cell culture substrates that mimic the structure and composition of the native ECM [40].

In this review, we discuss the use of biomaterials to develop 3D models of aging-related neurodegeneration, muscle degeneration, and neuromuscular junction disintegration. We first provide a brief overview of the types of biomaterials that have been developed for these purposes, emphasizing recent advancements in dynamic biomaterials. For a more detailed overview of biomaterial properties and their impact on 3D cell culture models, readers are directed to several recent review articles [41,42,43,44,45,46,47]. We subsequently provide a more detailed account of the use of such material-based approaches to model the neural and muscular systems in order to investigate the pathophysiology of aging.

## 2. Biomaterials as Models of the Aging Extracellular Environment

Biomaterials are broadly defined as any materials that interact with biological systems [48,49]. Metals, ceramics, and composites have been widely used in the manufacture of medical devices such as implants, tissue fixation devices, joint replacements, stents, or surface coatings [48]. More recently, polymers have been preferred over other types of materials for biomedical and pharmaceutical purposes, as polymers can be readily engineered to mimic the extracellular environment for the purpose of eliciting specific responses that recapitulate physiological processes in vitro or that have a therapeutic effect in organisms in vivo [48,50,51]. This is particularly the case for hydrogels, as their physio-chemical properties can be designed to be comparable to those of the native extracellular matrix (ECM), the polymeric meshwork that surrounds cells in vivo. Thus, hydrogels have become the preferred biomaterials for engineering 3D in vitro tissue models [52,53,54].

Hydrogel biomaterials are 3D crosslinked networks that can absorb a significant amount of water due to their hydrophilic composition [37,49,55,56]. Hydrogels are formed by connecting polymer chains with either covalent or non-covalent linkages [49,57]. Hydrogels can be classified as natural or synthetic depending on the origin of the polymers comprising the gels [49,58]. Natural hydrogels such as collagen and hyaluronic acid are obtained from living organisms and are thus biocompatible, bioactive, and biodegradable [55,58]. However, such naturally derived materials often have low mechanical strength and their compositions may be highly variable between different batches [49]. On the other hand, synthetic hydrogel biomaterials have a higher reproducibility in their physio-chemical properties due to the enhanced control of their structure and composition afforded by chemical synthesis [59]. In particular, chemical synthetic approaches allow for greater control over crosslinking chemistries, resulting in the generation of materials with superior mechanical properties in comparison with natural hydrogels [55]. Due to their synthetic nature, fully synthetic materials do not have bioactive moieties in their structure [55,60]. While this inherently results in less interaction between the materials and living cells, which may necessitate further functionalization to encourage desired bioactivity, the lack of interaction may result in lower immunogenicity for implanted materials [61,62,63].

Since the implantation of the first metal-based biomaterial in the 1800s, the field has moved from the use of bio-inert materials to the development of bioactive and biodegradable implants that direct cell behavior and promote the regeneration of the host tissue [56,57,64]. Further advancements made in polymer chemistry during the past decades have allowed the development of dynamic biomaterials, which detect physical or biochemical stimuli from their environment and respond to them by changing their structure and mechanical properties [56,57,64,65,66,67]. The use of these materials in culture systems results in an improved biomimicry of the cell microenvironment in comparison with static hydrogels, as the native ECM is constantly remodeled in response to biophysical stimuli or biochemical cues produced by cells during development, tissue homeostasis, or disease [40,66,68,69,70]. This is of particular importance in disease modeling as any alteration in the ECM structure or composition affects cell behavior and therefore may prove to be a key determinant of disease progression [69]. Analogously, the use of dynamic materials is particularly relevant in the research of aging-related pathologies due to their time-dependent properties that may mimic the temporal progression of the aging process [41]. Beyond their potential applications in modeling disease and aging in 3D culture systems, dynamic materials have been extensively studied in regenerative medicine applications, as such materials can be designed to integrate with the host microenvironment and evolve in response to physiological processes [71].

In the native cell microenvironment, cells interact with the matrix and other cells while also experiencing exogenous forces such as gravity and fluid shear stress [67]. Therefore, both cells and the ECM are subjected to endogenous and exogenous forces that have an effect on cellular responses such as differentiation, proliferation, and migration [67]. Since dynamic biomaterials can respond to cell-derived cues or external physio-chemical stimuli, or a combination of both, they are excellent candidates to replicate key aspects of the mechanical microenvironment of living tissues. Dynamic biomaterials can be classified, according to the type of stimuli that these materials respond to, as inherently dynamic (cell-responsive) or on-demand tunable systems [41].

Inherently dynamic biomaterials are remodeled through biochemical or biophysical interactions with living cells. The changes in material properties elicited by cells, such as altered stiffness or nanostructure, can subsequently cause changes in cell behavior [56,71,72]. Thus, a bidirectional communication that mimics the native ECM–cell dynamics can be achieved [66]. The synthesis of inherently dynamic materials may involve the creation of reversible crosslinks such as supramolecular host–guest complexes, hydrophobic interactions, hydrogen bonds, dynamic covalent networks, and ionic interactions, or the introduction of chemically responsive moieties such as hydrolyzable ester linkages or enzymatically degradable peptide sequences (Table 1) [66,73,74]. The introduction of bioresponsive motifs in the hydrogel networks confers the engineered matrices with the ability to undergo changes in mechanical and structural properties in the presence of biomolecules such as enzymes or upon exposure to an altered redox state or pH [41,67]. The development of proteolytically degradable hydrogels that can be remodeled and degraded by enzymes such as MMPs is of particular interest in aging research as it allows investigation of the changes in matrix proteolysis that occur with aging [41].

As mentioned above, dynamic biomaterials that respond to cell-derived cues capture a key aspect of the reciprocal interactions between cells and their microenvironment. However, such materials do not allow spatiotemporal control over the resultant changes in material properties and thus do not fully replicate the long-term, aging-related physiological changes in vitro. Therefore, the development of bioengineered disease models will benefit greatly from the use of dynamic materials that are tunable on demand upon application of external stimuli such as light, temperature, magnetic and electric fields, strain, and ultrasound (Table 2) [41,66,67,100]. These stimuli-responsive materials enable researchers to quickly generate substantial changes in the mechanical and biochemical properties of the materials, while also controlling which regions of the engineered matrices are subjected to such alterations, better recapitulating the dramatic and heterogeneous changes seen in aged tissues.

It will ultimately be important to combine aspects of both inherently and on-demand tunable dynamic materials to adequately capture the complexities of the aging microenvironment in vitro: first relying on inherently dynamic materials to enable cells to assemble into 3D cultures that resemble healthy tissue and then changing the matrix properties on demand to simulate an aged disease state [41]. In particular, research on aging-related neurodegeneration and muscle degeneration will benefit from dynamic biomaterials that replicate the changes in the mechanical properties of the ECM that occur with aging within a shorter time frame than that of native tissues. Furthermore, these engineered platforms can be adapted to mimic the mechanical stimuli that neural and muscular tissues receive while exercising or when exposed to fluid shear stress. The following sections review the advancements made in the bioengineering of 3D in vitro models of aging-related neural and muscular degeneration, with a particular focus on the use of dynamic materials to improve the biomimicry of these systems.

## 3. Modeling Neurodegeneration

Neurodegenerative diseases are defined by a progressive loss of neural cells and the synapses formed between them [118]. Alzheimer’s and Parkinson’s are the most common neurodegenerative disorders, and both are characterized by the formation of protein aggregates [119]. Extracellular amyloid plaques formed from amyloid β (Aβ) protein fragments and intracellularly aggregating neurofibrillary tangles (NFTs), composed mainly of tau protein, are formed in the brain of patients affected by Alzheimer’s, who present with memory loss and movement dysfunction [120,121]. The pathophysiological hallmark of Parkinson’s is the formation of intracellular α-synuclein inclusions, and the disease manifests with the loss of dopaminergic neurons in the substantia nigra. This causes bradykinesia, muscle rigidity, and a resting tremor in patients [118,119,122]. Due to increasing life expectancy, a rise in the prevalence of these disorders is predicted in the coming years. Beyond the detrimental impact that neurodegenerative diseases have on the quality of life of patients, they also pose a substantial burden to healthcare systems due to the intensive and lengthy care required [118].

The brain is made up of neurons and glial cells, which include astrocytes, microglia, oligodendrocytes, and ependymal cells [118,123]. A population of neural stem cells (NSCs) that give rise to both neural and glial cells is also present in brain tissue [124]. Constant communication between these different cell types is essential for proper brain development and function [125]. Furthermore, the interactions established between cells and the ECM are also crucial for the maintenance of neuronal function [126]. The brain ECM is essential for the maintenance of structural integrity and homeostasis as it contains cell adhesion motifs and growth factors that modulate neurogenesis, cell migration, axon guidance, and synaptogenesis [37,127,128,129]. This matrix constitutes 10–20% of the whole brain volume and presents components and features that are exclusive to this organ [130]. With a Young’s modulus of approximately 0.1–3 kPa, depending on the region analyzed, the brain is one of the softest tissues in the human body [37,118]. In comparison to the ECM of other organs, the brain ECM has a low number of fibrous proteins such as collagen and fibronectin, while glycosaminoglycans (GAGs) are present at a high concentration [37,131,132]. One of the most abundant GAGs in the brain ECM is hyaluronic acid (HA), which contributes to ECM integrity by establishing non-covalent interactions with other matrix components [37,133]. The rest of the GAGs are covalently linked to proteins to form proteoglycans (PGs) [37]. Heparan sulfate proteoglycans (HSPGs) and chondroitin sulfate proteoglycans (CSPGs) are the most abundant PGs in the brain, and their polarity confers brain tissue with a high water content of approximately 80% of its total mass [37,133,134]. Another major glycoprotein in the brain ECM is laminin, which regulates neural progenitor cell proliferation, neuronal survival, and synaptic activity [135]. The composition of the adult brain ECM is relatively stable, although it changes between different regions [136]. Despite this variability, any region of the brain consists of similar ECM structures, including a basement membrane, or basal lamina, that surrounds the blood vessels; perineuronal nets (PNNs) that can be found around the dendrites and neuronal cell bodies; and the interstitial matrix [130]. 

Aging and aging-related diseases cause alterations in the structure and composition of the ECM, which in turn affect the physical properties of neural tissue [37]. While the general trend is for brain tissue to become stiffer with age, the opposite effect is observed in Alzheimer’s disease patients [137]. These changes in the matrix are of particular importance in the context of neurodegenerative disorders as they modulate the accumulation and propagation of pathogenic molecules and have an effect on inflammation and neurodegeneration [9]. Neural and glial cells sense alterations in the mechanical properties of the matrix through mechanosensitive pathways that lead to alterations in cell behavior that can in turn contribute to disease progression. For example, HA remodeling has been observed in a mouse model of α-synuclein-induced dopaminergic neurodegeneration [37]. Such alteration in the ECM could contribute to the neurodegenerative events that are characteristic of this pathology [138]. 

Owing to the likely effects of changes in the ECM on the cellular pathophysiology of neurodegeneration, research on such disorders benefits from the development of 3D models that mimic the ECM, particularly those that are comprised of materials that replicate the dynamic properties of the native matrix [9,118]. Such systems will also contribute to an improved comprehension of the mechanisms of Aβ aggregation in the brain ECM, since 2D models do not facilitate the formation of Aβ plaques or neurofibrillary tangles due to the dilution of the constituent proteins in the media [9]. On the other hand, 3D culture conditions have been reported to facilitate the formation of amyloid plaques within 6–12 weeks [139]. Therefore, the use of these models increases our understanding of the pathological features of amyloid disorders such as Alzheimer’s [9]. As the formation of Aβ agglomerates and subsequent fibrillogenesis are affected by the physio-chemical properties of the interstitial fluid and the biopolymer fibers that form the ECM, it is essential to consider these conditions when formulating synthetic matrices and the buffers used in biofabrication processes [9]. The replication of the brain ECM microenvironment will be extremely valuable for the understanding of the potential effect of the matrix on the accumulation of Aβ and tau proteins in the interstitial fluid, since the ECM exerts a resistive force to fluid flow and clearance and may lead to the generation of hydrostatic and osmotic pressure [140,141]. In addition, the ECM restricts the transport of Aβ oligomers and tau proteins across the brain [142,143]. Thus, the replication of disease- and age-related matrix alterations is an essential requirement to mimic and understand protein aggregation in amyloid neurodegenerative disorders [144,145,146,147]. Besides the improved comprehension of the effects of physical structure and phenomena on neurodegeneration, the use of 3D models facilitates the observation of cellular responses that only occur in an environment that is comparable to the native cell niche [37]. For example, observations made in vivo have shown that astrocytes can switch from a phenotype with neuro-supportive characteristics to one that is neuro-toxic [148]. This phenomenon cannot be replicated in a 2D culture due to the high reactivity of astrocytes in 2D conditions. On the contrary, the use of collagen matrices maintains astrocytes at reduced reactivity, hence allowing the activation of reactive states in response to physiologically relevant cues [149,150].

The development of 3D in vitro models of neurodegeneration has been made possible by the breakthroughs made in different disciplines such as stem cell biology, genetic engineering, cell reprogramming, and biomaterials science [37]. Human pluripotent stem cells are a promising tool in the development of 3D models as they eliminate the need to use animal cell lines and facilitate the generation of patient-derived neural cells, enabling studies of cells that may be genetically predisposed to developing neurodegenerative disorders [131]. Organoids derived from these stem cells have also emerged as a tool with huge potential to study organ development and disease [151,152]. However, these systems are often arrested at an embryonic developmental state, and it is not yet clear how to mature these organoids to make it possible to model the aged brain [9]. In parallel to the advancements made in cell and molecular biology, a wide variety of ECM-like materials has been developed to support cell growth in 3D while providing biomimetic mechanical, topological, and biochemical cues [153,154]. Furthermore, the development of dynamic biomaterials may facilitate the reciprocal cell–matrix interactions that occur in the cell microenvironment [37]. Beyond considerations of biocompatibility and bioactivity generally required for cell culture, the use of electrically conductive biomaterials may hold particular promise for the culture of neural cells due to the positive effect of electrical stimulation on neuronal differentiation [155].

Hydrogels are the most commonly used biomaterials in neural cell culture as they form soft and porous networks that resemble the brain ECM and can be modified to present biochemical and biophysical cues characteristic of brain tissue [156,157]. As mentioned in previous sections, these materials are composed of polymers that are either natural or synthetic. Matrigel^®^ and decellularized ECM are some examples of natural hydrogels that have been widely used in neural cell culture [158,159]. The former has been used by Kim and coworkers to produce a 3D scaffold where neurons expressing AD mutations were cultured. The resulting model reproduced the aggregation of amyloid-β and the accumulation of hyperphosphorylated tau [139]. However, a problem associated with the use of Matrigel^®^ in neural cell culture is that, as a murine sarcoma-derived matrix, it lacks numerous ECM components that direct brain development [160]. Furthermore, Matrigel^®^ contains growth factors that may affect neural cell physiology [161]. For studies or applications where only specific components of the matrix are needed, such as collagen, HA, or laminin, these can be purified or produced as recombinant proteins [37,162]. Collagen has been used in the production of 3D cell culture systems due to the proven efficacy of this protein in promoting cell survival in vitro [163]. Liu et al. used this material to encapsulate mast cells, an innate immune cell type found in the proximity of Aβ aggregates, in the presence of different Aβ peptides. The authors found that these conditions caused mast cells to secrete inflammatory mediators and hence concluded that the 3D collagen model is a valuable tool for investigating the mechanisms of AD-related neuroinflammation [164]. Laminin is often incorporated in 3D brain tissue models due to the ability of this protein to promote neural cell adhesion and drive neurite outgrowth [165,166,167]. In an example of the use of this protein to improve the bioactivity of bioengineered models of neurodegeneration, Cantley et al. reported the development of a 3D human model that promotes the differentiation and long-term culture of functional neural networks generated from hiPSCs derived from either healthy individuals or patients affected by Alzheimer’s or Parkinson’s diseases. The fabrication of this model involved the formation of a silk fibroin scaffold that was filled with collagen and coated with poly-L-ornithine and laminin to promote 3D growth. The resulting matrix supported the direct codifferentiation of hiPSCs into neurons and astrocytes. Electrophysiological recording and calcium imaging demonstrated the functional activity of the 3D brain tissue model for nine months [168]. Therefore, this model could be a valuable tool for the investigation of network development, maturation, and degeneration, as well as for the discovery of early-stage biomarkers of neurodegeneration. Whilst ECM-derived proteins are superior to other materials in terms of bioactivity, the challenges encountered in their isolation and characterization have promoted the use of alternative biopolymers by some researchers. Polysaccharides such as chitosan, alginate, or agarose, which are mainly obtained from marine organisms, have been widely used to fabricate scaffolds for tissue engineering applications [169,170]. Tedesco and coworkers reported the use of chitosan to fabricate a microbead-based scaffold that supported the growth of functional 3D neuronal networks [171]. Microbead scaffolds have also been produced with alginate to generate a model of Alzheimer’s disease that contained amyloid-secreting cells. The adjustment of the protocol to produce beads with different sizes and stiffness allows the formation of highly tunable and biomimetic structures [172].

Despite the promising developments made in the generation of 3D brain models using natural hydrogels, it is important to take into consideration the batch-to-batch variation and poor control over the mechanical properties of these materials. Therefore, the use of synthetic hydrogels such as poly(ethylene glycol) can be particularly advantageous for studies where a specific parameter needs to be controlled in order to investigate its effects on cell behavior [37]. Furthermore, 3D models made with synthetic materials have been shown to support the production of pathogenic Aβ peptides and phosphorylated tau proteins, as demonstrated by Ranjan et al. using PLGA microfibers seeded with iPSC-derived neural progenitor cells (NPCs) with Alzheimer’s disease mutations [173]. While some of the challenges encountered in 3D modeling in vitro can be overcome with the use of synthetic hydrogels, these present other pitfalls such as a low bioactivity due to the absence of ECM-derived motifs. Nevertheless, the development of synthetic self-assembling peptides represents a propitious advancement in the field, as these materials circumvent the issues with mechanics and bioactivity encountered with natural and synthetic materials, respectively. Self-assembling peptides (SAP) are particularly interesting materials since they allow the rational design of synthetic hydrogels using natural amino acids as building blocks that can be combined in multiple ways to generate specific architectures and bioactive sequences [174]. The most widely used peptide in neural tissue engineering is a four-amino acid sequence composed of arginine-alanine-aspartate-alanine (RADA) [175,176,177]. This system has been used by Ni and coworkers to produce RADA16-I hydrogels that improve the differentiation of dopaminergic neurons from murine iPSCs and embryonic stem cells (ESCs), hence providing a model for the investigation of Parkinson’s disease [177]. However, despite the careful design of synthetic hydrogels to mimic the brain ECM, their low stiffness often results in structural instability that compromises the durability of the model in vitro. This issue can be ameliorated by including a second polymer network in the hydrogel in order to improve the mechanical stability [37]. Scaffolds made of silk or other fibrous polymers produced by electrospinning are some examples of fibrous materials used for this purpose in the development of 3D neural tissue models [168,178,179,180,181]. In addition to the advantages offered by synthetic materials in terms of control over their mechanical properties, they also provide additional functions such as electroconductivity. An example of such a conductive material is poly(3,4-ethylenedioxythiophene) (PEDOT), which can electrically stimulate bioengineered neural tissues [182].

The replication of the complex architecture of brain tissue often requires the use of biofabrication methods to convert the biomaterials into biomimetic models that are representative of the native cell environment. One of these techniques is bioprinting, an additive manufacturing method that allows the generation of cell-laden 3D structures with a predetermined design by depositing a cell-loaded biomaterial layer by layer [183,184]. Bioprinting technology facilitates the production of 3D matrices with multiple materials and cell types, as these different components can be loaded into separate print heads. The use of this technique to print neurons and glial cells in the same system allows the generation of complex models that can be used to study the interactions between the different cell types. Furthermore, a high degree of control over the architecture and cell position is achieved. Joung et al. harnessed this technique to develop a 3D model of damaged central nervous system (CNS) composed of iPSC-derived spinal neurons and oligodendrocytes [185]. The cell clusters were deposited at specific locations in the matrix through extrusion-based multimaterial 3D bioprinting. Furthermore, the oligodendrocytes had the ability to myelinate the axons. Other studies have also confirmed the suitability of this technique to generate models of Alzheimer’s and Parkinson’s disease. Benwood et al. differentiated healthy and patient-derived iPSCs into NPCs, and incorporated these cells into dome-shaped constructs that promoted the differentiation of NPCs into basal forebrain-resembling cholinergic neurons (BFCN), the first cell type affected in the progression of Alzheimer’s disease [186]. With regard to the application of this technology in the production of models of Parkinson’s disease, Abdelrahman et al. reported the fabrication of a peptide-based model that maintained the activity of encapsulated dopaminergic neurons for more than one month. Furthermore, co-culture with endothelial cells led to vascularization, which promoted neurite outgrowth and thus resulted in the formation of a dense neural network [187]. The vasculature improves the physiological mimicry of the system and improves the viability of the cultured cells due to the improved nutrient distribution. This is of particular importance when replicating the pathophysiology of neurodegenerative disorders due to the long time scales of disease progression [188].

Another technique that has revolutionized the development of in vitro models of brain tissue is microfluidics. The use of this technology has given rise to the creation of brain-on-a-chip models that allow compartmentalization and the application of fluid flow [189], which is of particular importance to investigate the effect of the shear stress generated by interstitial flow on neural cells [118]. The design of these devices allows the use of hydrogel matrices to culture the cells in 3D [190]. In addition, the transparent materials used in the fabrication of these platforms make it easy to obtain images of the cells with optical microscopy techniques [191,192]. An example of the use of this technology to replicate neurodegenerative diseases is the three-compartment organ-on-a-chip device developed by Virlogeux et al. to model the Huntington’s disease (HD) corticostriatal network. Each of the three chambers corresponds to either the presynaptic, synaptic, or postsynaptic region [193]. A similar system has also been used by Li et al. to investigate the neurotoxicity of the Aβ peptides that are characteristic of Alzheimer’s disease [194]. In another model of this disorder, Park et al. incorporated iPSC-derived neurons, astrocytes, and microglia cells in a microfluidic platform to replicate beta-amyloid aggregation, neuroinflammatory activity, and phosphorylated tau accumulation [195].

As a whole, the studies described above demonstrate that the use of bioprinting and organ-on-a-chip technologies can enable researchers to create 3D models with complex structures that resemble the native ECM architecture. Furthermore, the use of microfluidic systems facilitates the replication of fluid shear stress effects in vitro. These technologies can also be combined to increase the automatization of the resulting models and to achieve a higher precision and resolution when depositing the bioinks [196,197].

## 4. Modeling Aging-Related Muscle Degeneration

While many of the aging-related neurodegenerative disorders discussed above manifest with impaired motor control, aging also results in substantial changes to the skeletal muscle microenvironment that can result in muscle loss and directly cause impaired mobility. Skeletal muscle is a highly organized tissue formed of several bundles of fascicles that are in turn composed of muscle fibers, known as myofibers [198]. Two types of muscle fibers, type I (slow-twitch) and type II (fast-twitch), coexist in muscle tissue [199,200]. Each multinucleated myofiber is composed of multiple myofibrils [201], which in turn contain a series of aligned sarcomeres [198,202]. The muscle sarcomere is the basic contractile unit of the muscle fiber [198], and it is composed of thick and thin filaments of myosin and actin, respectively, which are perpendicularly linked to the Z-lines that are found on either side of the sarcomere [202,203]. When a muscle fiber receives a contraction signal through the neuromuscular junction, the myosin head binds the actin filament, forming a cross-bridge [198]. Consequently, the myosin head moves and slides the actin filament toward the middle of the sarcomere (M-line) (Figure 2) [204,205]. The contractions of numerous sarcomeres in series cause the contraction of the myofibrils, which in turn causes a shortening of the muscle fiber [204].

Each of the constitutive layers of muscle tissue described above is surrounded by the extracellular matrix [200]. This structure provides support to muscle fibers, nerves, and blood vessels, and it also facilitates the communication between myofibers and other cell types such as myoblasts, fibroblasts, and inflammatory cells [206,207,208]. Furthermore, it plays a key role in the regulation of satellite cells, also known as muscle stem cells (MuSCs), which differentiate into myogenic progenitor cells that form new myofibers during muscle repair [30,200,209,210]. Depending on the muscle layer that it covers, the skeletal muscle ECM is traditionally classified as epimysium (encasing the entire muscle), perimysium (encasing groups of muscle fibers, or fascicles), and endomysium, or basal lamina (encasing the muscle fibers) [200,211,212]. However, microscopic observations of muscle tissue have shown that this classification may be relatively arbitrary and that the endomysium and perimysium form two continuous 3D networks [211,213]. Furthermore, according to some classification schemes, the basal lamina is considered to be a distinct structure from the endomysium; however, they are in close contact, and in some instances, the basal lamina has been described as a specialized form of endomysium [214,215]. Generally, the major component of muscle ECM is collagen, accounting for up to 10% of its dry weight [200,211]. Other constituent elements of this matrix include glycoproteins, proteoglycans, and elastin, although variations in the composition have been reported between the different ECM layers [216,217]. The epimysium is formed of collagen type I, collagen type III, and fibronectin; the perimysium contains fibronectin, proteoglycans, and collagens I, III, V, and VI; and the endomysium is composed of laminin, fibronectin, and collagens I, III, and V [218,219,220,221]. Collagen IV, the collagenous component of basement membranes, has also been isolated from the endomysium, where it links with laminin to form a complex involved in force transduction [214,218]. Some studies suggest that collagen type I is the main component of intramuscular ECM and that it is abundant in the perimysium and epimysium, while collagen type III is evenly distributed between the endomysium and the epimysium [214,218,222]. However, further investigation with standardized models is required to reach a consensus on the muscle matrix composition and to determine whether collagen-type ratios vary between muscles with different functions [211].

During aging, the skeletal muscle ECM undergoes fibrotic ECM deposition, a reduced collagen turnover, and an increase in non-enzymatic crosslinking of collagen fibers through the accumulation of advanced glycation end-products (AGEs) [223,224,225]. These changes result in an increase in muscle stiffness that renders this tissue more susceptible to injury [216,226]. The ECM architecture is also affected by aging, with a replacement of the ordered crisscross lattice structure of healthy perimysium by a disordered fiber network [213,217]. Results from numerous studies indicate that the absolute collagen content increases and that the collagen composition of the basal lamina changes with age [217,227,228]. Specifically, a reduction in the amount of collagen type IV in the basal lamina has been observed in older humans, while the laminin content increased [229]. It is important to note that opposite trends have been observed when using animal models to analyze these age-related changes [228]. Due to the contact between the basal lamina and the resident muscle stem cells, these changes in the ECM mechanics and composition may compromise the functionality of the stem cells and alter their biochemical communication with myofibers [230,231]. Furthermore, the impaired degradation of ECM proteins such as fibronectin, elastin, and laminin leads to an accumulation of proteolytic fragments that contribute to the reduced adhesion between MuSCs and myofibers [232]. Therefore, the regenerative capacity of skeletal muscle may be negatively affected by the age-related ECM alterations described herein [216,230].

Age-related changes in muscle structure result in the degeneration of this tissue, a condition known as sarcopenia. This disorder is defined as the loss of muscle mass and function that occurs beginning from middle age [233,234,235]. This syndrome results in physical frailty and an increased risk of developing chronic diseases, as well as a loss of independence in patients and a tendency toward sedentarism [234,236]. The detrimental effects of sarcopenia on overall health are due to the loss of the ability of muscle to generate force and to regulate metabolism throughout the body [234]. In addition to suffering from muscle degeneration directly related to aging, the elderly population is also at a higher risk of suffering from cachexia, which is the loss of muscle mass due to underlying pathological conditions such as cancer, heart failure, pulmonary disease, HIV, and renal and liver failure. Due to the higher incidence of these conditions among the elderly, this age group is particularly vulnerable to cachexia [237].

On a cellular level, the two mechanisms that regulate the decline in muscle mass are muscle fiber atrophy and muscle fiber loss, with type II fibers more affected than type I counterparts [233,234,238]. These changes are linked to an imbalance in muscle protein synthesis and degradation. The disproportionate increase in protein degradation is caused by muscle denervation, dysregulation of proteasomal degradation pathways, mitochondrial dysfunction, intramuscular and intermuscular fat infiltration, and inflammatory and hormonal changes [235,239,240]. These factors also result in a decline in the number and function of satellite cells, also known as muscle stem cells (MuSCs), in sarcopenic muscle [233,241,242,243,244]. This can be due to alterations in the systemic factors that regulate MuSCs, which include MuSC niche factors, TGF-β, and myostatin [235]. Other contributors to muscle degeneration include neuromuscular junction dysfunction (discussed below), reduced number of motor units, inflammation, insulin resistance, and oxidative stress [245,246,247,248,249,250].

The management of sarcopenia is based on physical activity and dietary adjustments such as an appropriate intake of protein, vitamins, antioxidants, and polyunsaturated fatty acids [235,251]. However, no pharmacological agents are currently available for individuals who do not show any signs of improvement [251]. Therefore, the development of novel therapies to combat muscle degeneration is currently a pressing need. To achieve this, it is of paramount importance to increase our understanding of the pathophysiological mechanisms of muscle wasting. Since most of our current knowledge in this area has been acquired through studies based on 2D culture systems or in animal models that do not faithfully recapitulate human aging, this field would benefit from the use of bioengineered 3D models of aged human muscle tissue.

The generation of skeletal muscle tissue in vitro requires the integration of multiple cell types and the recreation of the extracellular environment and tissue architecture. The latter is a particular challenge in muscle tissue engineering due to the complexity of its hierarchical organization [214]. While the hallmark cell types in skeletal muscle tissue are myofibers and muscle stem cells, other crucial cell populations that can be found in muscle include fibroblasts and fibro-adipogenic progenitors, pericytes, endothelial cells, neural cells, and immune cells. Therefore, the incorporation of different cell types into engineered skeletal tissue models has been shown to improve the models’ fidelity and the maturation of the tissue constructs in vitro [214]. Due to the ability of iPSCs to differentiate into multiple cell types, these cells are widely used in the development of 3D models of skeletal muscle [252]. For instance, Maffioletti et al. employed iPSCs derived from patients suffering from Duchenne muscular dystrophy to generate 3D muscle models based on hydrogels subjected to uniaxial tension in order to increase the degree of myofiber alignment [253]. Stem-cell-derived muscle cells can also form aggregates, known as myospheres, which enable scaffold-free 3D culture with cell–cell interactions that closely resemble those of native muscle [254,255]. With regard to the muscle ECM, it is essential to consider the mechanical characteristics of this structure as they influence cell behavior and therefore play a major role in muscle development and regeneration [198]. Furthermore, this matrix provides structural support and mediates force transmission [256]. As the main component of the muscle ECM is collagen, this material has been used in the development of numerous muscle models [214]. The addition of Matrigel^®^ to collagen hydrogels improves the engineered muscle structure as it leads to greater force generation and muscle differentiation [257]. A study conducted by Wang and coworkers reported the creation of a 3D muscle aging model by culturing primary muscle progenitor cells in hydrogel matrices composed of Matrigel^®^ and fibrinogen. By culturing cells derived from either younger or older animals, they achieved the generation of a model that could show a sarcopenic phenotype in old-cell-derived constructs and a healthy tissue in the model derived from young cells. The sarcopenic phenotype was characterized by the presence of hypotrophic myotubes, reduced contractility, and impaired regenerative capacity [258]. While Matrigel^®^ has been proven suitable for the generation of artificial muscle in vitro, the lack of muscle-specific cues in this substrate makes it desirable to develop alternative materials that possess muscle-derived proteins [259]. Therefore, decellularized muscle ECM hydrogels are increasingly used in muscle tissue engineering as they contain growth factors and cell adhesion proteins that are essential to achieve the bioactivity required in a tissue model [260,261].

Owing to the importance of surface topography on cell alignment and function, the complex architecture of skeletal muscle must be replicated when developing 3D models of this tissue [262]. One approach to achieving these structures makes use of biofabrication techniques that enable the generation of biomimetic topographies on the culture matrices [214]. Such techniques include micropatterning, 3D bioprinting, and electrospinning [263]. Micropatterning techniques such as soft lithography, photolithography, or hot embossing lithography have been used to investigate the effect of different topographies such as grooves, ridges, channels, or posts on cell arrangement and function [264,265]. Charest and colleagues made use of hot embossing lithography to generate different micropatterns, such as grooves, ridges, and holes, with different sizes. The alignment and differentiation of myoblast cultures on such surfaces were analyzed and compared. The results of the study showed that a higher degree of alignment was achieved on narrower grooves [266]. Lithography technologies have also been used to generate models of muscle disease. Fernandez-Garibay et al. reported the utilization of these methods to create a bioengineered model of myotonic dystrophy type 1. Gels composed of gelatin methacrylate (GelMA)-carboxymethyl cellulose methacrylate (CMCMA) were micropatterned with a topography that promoted myoblast alignment and differentiation. Patient-derived cells cultured on these structures led to the generation of aligned myotubes with disease-associated characteristics [262].

Lithography-based technologies are particularly advantageous for the generation of bioinspired topographies due to their versatility, reproducibility, and high throughput [267]. However, they also present several pitfalls, such as the difficulty in generating certain designs such as closed loops, the deposition of residual chemical agents, and the long time required to make design changes in such multi-step processes [268,269]. While UV photolithography has been presented as an alternative to traditional lithography, this process requires a photomask and long processing times [269]. On the contrary, laser ablation is completed in a single step and without any additional materials or chemicals. To minimize the thermal effects associated with laser machining, femtosecond laser ablation has emerged as a technique that can generate specific patterns on biological substrates under physiological conditions [269,270]. However, these technologies pose limitations on the scaffold size and, consequently, make it difficult to produce a whole organ or a large tissue [271]. To circumvent these issues, bioprinting technology based on the layer-by-layer fabrication of digitally designed structures has emerged as a promising alternative to the use of physical templates [272]. This technique allows the generation of complex geometries with a precise placement of cells, materials, and biomolecules [264]. Therefore, a higher accuracy in the replication of native tissue and an improved reproducibility are achieved [214]. This is of particular importance in muscle tissue engineering due to the complex organization of this tissue [254]. The selection of a suitable bioink with an appropriate printability, biocompatibility, mechanical strength, and degradation rate is essential when using this technology [264]. Furrer et al. showed how Matrigel^®^ can be combined with 3D bioprinting technology to obtain human skeletal muscle models [214]. Matrigel^®^ has also been used in combination with fibrinogen to 3D print a model of aged human skeletal muscle. This platform supported the differentiation of myoblasts into functional myofibers that showed signs of aging after treatment with TNFα to investigate the morphological and functional changes that are associated with this process [273]. Due to the issues encountered with the use of Matrigel^®^, other materials such as decellularized ECM have been explored for 3D bioprinting applications [261]. Moreover, higher levels of expression of muscle differentiation-related genes MYF5, MYOG, MYOD, and MYHC were reported in cells cultured on decellularized ECM [264]. High yields of myotube formation have been reported when using this material to culture myoblasts [274]. 

The generation of functional muscle microtissues is only one part of the modeling process when replicating certain types of muscle degeneration in vitro. This is the case of cachexia, the loss of muscle tissue due to the incidence of other diseases such as cancer. Due to the involvement of different organs in this disorder, the development of models of cachexia requires the integration of different cell types and ECM-mimicking biomaterials. Therefore, the use of bioprinting technology is of particular relevance in these studies, as these systems are designed to incorporate different materials and cells into the printed construct. As such, Garcia-Lizarribar et al. made use of this technique to generate functional skeletal muscle that was cultured in a cancer-cell-conditioned medium to replicate the effects of cytokines produced by melanoma cancer cells on muscle tissue [275]. An often overlooked parameter in sarcopenia or cachexia modeling is the influence of exercise on disease progression and therapeutic outcomes. However, recent studies are starting to replicate these effects in engineered muscle models in vitro. As shown by Reyes-Furrer et al., bioprinting technology can be used to generate advanced models with specific designs that facilitate the electrical stimulation of the resulting constructs to mimic the effects of exercise-induced muscle contraction [259].

Aligned nanofibrous materials have been shown to be superior to micropatterned materials in terms of the assembly of longer myotubes [276]. Electrospinning has long been recognized as a powerful technique for the production of micro- and nanofibers [277,278]. Its popularity in the field of tissue engineering stems from its versatility, low cost, and ability to generate nanoscale topological cues comparable to those of the ECM [278]. Aligned fibers can be generated with this technique, hence enabling the replication of the muscle tissue architecture. The result of using such architectures in the generation of muscle models is a more efficient organization and alignment of myotubes [279]. Soliman et al. harnessed this principle to engineer a muscle sheet model for studies of the neuromuscular junction in vitro. Muscle progenitor cells cultured on aligned fibers maintained directionality to a higher extent than cells grown on traditional culture surfaces [280]. While numerous studies have proven the ability of electrospun nanofibers to modulate cell organization, the infiltration of cells into the dense fiber mesh remains a challenge encountered with this technique. Fibers produced with the conventional electrospinning technique form 2D mats rather than 3D structures. Therefore, the formation of 3D nanofibrous scaffolds requires the use of 3D near-field electrospinning or melt electrowriting (MEW) technology [281,282].

Besides the selection of the appropriate cells, biomaterials, and biofabrication techniques, another factor that is essential to consider when engineering muscle tissue in vitro is the application of electrical and mechanical stimulation. Such stimulation improves cell proliferation, maturation, and alignment. Bioreactors and deformable biomaterials have been developed to mimic the effects of physical exercise on muscle tissue [214]. Numerous studies have corroborated the beneficial effects of mechanical stimulation, such as cyclic loading or stretch, on the generation of 3D muscle models, as such stimuli promote ECM remodeling, hypertrophy, alignment, and satellite cell activation and differentiation [254,283,284,285,286,287]. Furthermore, the emergence of organ-on-a-chip technology facilitates the application of mechanical stimuli such as fluid shear stress or hydraulic compression on muscle microtissues [288,289,290]. Electrical stimuli have also been applied to 3D muscle models [291]. Long-term electrical stimulation of primary myoblasts cultured on hydrogels was shown to enhance myotube formation and a higher differentiation efficiency [292]. Therefore, conductive materials such as carbon nanotubes, PEDOT, or gold nanoparticles have been incorporated into many scaffold-based models [293,294,295,296].

## 5. Modeling Aging-Related Neuromuscular Junction Degeneration

The nervous system and skeletal muscle are connected via neuromuscular junctions (NMJs). The NMJ is a synapse specialized in the transmission of electric impulses from the motor neuron to the innervated muscle fibers [198]. It is composed of three elements: pre-synaptic (motor nerve terminal), intrasynaptic (synaptic basal lamina), and post-synaptic (muscle fiber and muscle membrane) (Figure 3) [19]. When an action potential reaches the pre-synaptic element, voltage-dependent calcium channels open and allow calcium to enter the neuron, triggering the release of acetylcholine (ACh) into the synaptic cleft. ACh diffuses rapidly through the basal lamina, the ECM of the synaptic cleft, to bind its postsynaptic receptors (AChRs), triggering a cascade of molecular events that leads to the contraction of the muscle fiber [20,198]. Briefly, these molecular events involve the generation of an action potential by the AChRs and the subsequent activation of voltage-gated dihydropyridine receptors (DHPRs) from the sarcolemma. This triggers the release of calcium from the sarcoplasmic reticulum through ryanodine receptors (RyRs) [297]. Then, calcium binds troponin C and causes a conformational change that moves tropomyosin away from the myosin-binding site present in actin filaments. This allows the formation of a cross-bridge between actin and myosin, triggering muscle contraction [298]. 

The intrasynaptic region of the NMJ is divided into the primary cleft, which extends from the presynaptic membrane to the basal lamina that encases the muscle fiber, and the secondary clefts, which includes the space between the junctional folds of the postsynaptic membrane [198]. The central region of the synaptic cleft contains the synaptic basal lamina, which differs in its composition from the extrasynaptic basal lamina [299]. This matrix maintains a tight adhesion between the pre- and post-synaptic elements and is involved in NMJ innervation, development, and regeneration [300]. The synaptic basal lamina is composed of a protein network composed of laminin (isoforms 221, 421, and 521) and collagens IV and XIII, which are interconnected by the proteoglycans agrin and perlecan and the glycoprotein nidogen-2 [198,298]. Synaptic laminins contribute to AChR clustering through interactions with their post-synaptic receptors: α7β1 integrins and dystroglycan. Agrin is also involved in AChR clustering through the activation of Lrp4 and MuSK. Furthermore, laminin 421 may mediate degenerative changes caused by aging [20].

A substantial proportion of the matrix components described above are subjected to alterations during aging. Consequently, cell–cell and cell–matrix interactions undergo a remodeling process that results in age-related alterations in NMJ morphology, mechanical properties, and molecular composition and function [301]. The pre- and post-synaptic membranes are particularly affected by morphological alterations, showing changes in topography. All these processes, together with inflammation, mitochondrial dysfunction, and oxidative stress, contribute to the degeneration of the NMJ, which may be a cause of the muscle degeneration associated with age [19]. However, whether NMJ degeneration is a cause or a consequence of sarcopenia is still to be resolved and it is a question of particular interest in the field [302].

Due to the difficulty of studying the NMJ in vivo, the development of 3D models of this synapse is of paramount importance to investigate the mechanisms of NMJ degeneration [303]. The implementation of such engineered models holds great promise in studying the pathophysiological mechanisms of diseases associated with the NMJ, such as amyotrophic lateral sclerosis (ALS). The use of iPSCs to generate neural and muscular tissues facilitates the generation of patient-derived NMJ models to elucidate the pathophysiological hallmarks of NMJ degeneration that are specific to each patient [304]. Recent developments in the field have proven fruitful with the creation of compartmentalized models that recreate both the neuronal and the muscular components of the NMJ. When neurons and muscle cells are cultured in proximity, the axons of neurons extend into the myofibers and form the NMJ [304]. A representative example of such a system is the model developed by Osaki et al., which consists of a compartmentalized platform comprised of iPSC-derived motor neurons from patients in the form of spheroids and skeletal muscle bundles. NMJs were successfully formed between the resulting microtissues and showed characteristics of ALS pathology such as motor neuron degradation and increased apoptosis of the muscle on the ALS motor unit [305]. Another NMJ model based on a co-culture was developed by Massih and coworkers. Briefly, primary myoblasts, fibroblasts, and iPSC-derived motor neurons were co-cultured in a hydrogel composed of Geltrex™, fibrinogen, and thrombin. By using iPSCs derived from ALS patients with *SOD1* mutations, the resulting motor neurons showed a decrease in muscle contractility over the 21 days of culture [306]. A challenge encountered in the generation of stem-cell-derived neuromuscular models is the detachment of muscle fibers from their culture substrate as they contract. To circumvent this issue, Kamm and colleagues created a culture system based on the support of myofibers between micropillar cantilevers that deflected upon contraction [305,307,308]. 

Microfluidic technology is particularly advantageous for the generation of compartmentalized systems, as it allows communication between the pre- and post-synaptic chambers and the production of customized arrangements that better mimic the anatomical characteristics of the NMJ. As previously mentioned, microfluidic platforms have distinctive advantages over other systems with regard to mechanical, biochemical, and electrical stimulation [309,310]. Furthermore, the observation of the nerve impulse transmission is facilitated by the transparency of the materials used in the manufacture of these organ-on-a-chip platforms [304]. This feature enables the integration of sensors to monitor the cellular response to biophysical stimuli and drugs [311]. Park and coworkers produced one of the first microfluidic-based NMJ models using mouse embryonic stem cells to generate the motor neurons and C2C12 myoblasts to produce the myofibers [312,313]. This work facilitated the development of more advanced models with physical separation between the pre- and post-synaptic compartments, hence supporting the formation of functional NMJs [308]. Later studies have shown that the addition of other cell types to the system, such as endothelial cells or astrocytes, leads to further improvement in NMJ development, functionality, and biomimicry [313,314,315]. The interactions between the different cell types are supported by using hydrogels as substrates, and the addition of extra chambers opens the possibility of modeling the interactions between the NMJ and other organs [314]. Different channel designs such as funnel-shaped systems have also been explored in order to improve axon pathfinding [316]. As demonstrated by Southam et al., the generation of alternative designs allows the replication of the anatomical features of the NMJ [314]. The creation of different designs can also improve the functionality of the model by facilitating cell seeding, as is the case for open-architecture microfluidic chips, or by controlling the culture media composition in different chambers [309,317].

Another technique that is particularly suitable for the generation of 3D cell culture compartments with different designs is bioprinting. This technology allows the generation of complex designs that better resemble the native architecture of tissues and organs. Recent developments have resulted in the generation of an NMJ model generated with C2C12-derived myotubes and motor neurons derived from mouse embryonic stem cells. Each of these cell types was cultured in ECM-derived proteins to create a ring-shaped muscle tissue with motor neuron embryoid bodies (EBs). The resulting structures could be combined and integrated with each other in order to form a multi-compartment system. The presence of glial cells in the EBs contributed to the survival, growth, differentiation, and proliferation of motor neurons and the subsequent improvement in NMJ maintenance [318,319]. Overall, all these studies show that great advancements can be achieved in the replication of the NMJ in vitro by combining the advantages provided by co-culture, compartmentalized organ-on-a-chip systems, and bioprinting. It is expected that further efforts in this field will lead to the generation of NMJ models with improved functionality and biomimicry, and that this will enable researchers to obtain greater insights into the effect of physical forces and biochemical factors on age-related NMJ degeneration.

## 6. Conclusions and Future Directions

Aging is a complex process that leads to degeneration and functional impairment across the whole organism. Aging-related degeneration of the neural and muscular systems has a major impact on patients’ health and quality of life due to the role of these tissues in cognition, movement, and a range of other bodily functions. Owing to the complexity of the aging process, investigating the underlying mechanisms of neuromuscular aging is particularly challenging. Furthermore, the widespread use of animal models has limited the translatability of our knowledge due to the anatomical and physiological differences that exist between animal and human organisms. The use of animal models in aging research is also associated with other challenges such as ethical considerations, extended experimental timelines, and higher costs associated with these long-term experiments. Bioengineered 3D models composed of human-derived cells are a promising alternative that can overcome these issues. This approach is based on the combination of cells and ECM-mimicking materials forming a 3D structure that resembles the native tissue. While naturally derived cell-interactive materials have been widely used in neural and muscular disease modeling, these endeavors would benefit from the application of dynamic materials that respond to cell- or user-generated stimuli to better capture the dramatic temporal changes in the tissue microenvironment that accompany aging and disease. Furthermore, engineered synthetic materials may be able to overcome the limited ability to tune the mechanical properties of naturally derived materials. However, such synthetic materials typically lack the bioactive cues that are present in ECM-derived polymers. A promising alternative that can provide both bioactivity and customizable mechanical properties is the development of recombinant protein-based biomaterials. Such materials can be endowed with chemical sequences that allow their responsiveness to biochemical and physical cues from their microenvironment, as recapitulating aging-associated phenomena will likely require materials that can respond to both cell-secreted factors and externally applied stimuli. The combination of novel dynamic materials with the use of biofabrication techniques such as bioprinting or electrospinning may open the door to the generation of complex biomimetic structures with dynamic functionality that mimic the native ECM remodeling observed in vivo. Finally, the application of organ-on-a-chip technology may enable the generation of models with improved biomimicry and improved control over other important system parameters, such as fluid shear stress and nutrient transport. Such systems are also advantageous for the real-time observation of cellular responses and the application of mechanical and electrical stimuli to the 3D cell culture models.

## Figures and Tables

**Figure 1 biomolecules-14-00069-f001:**
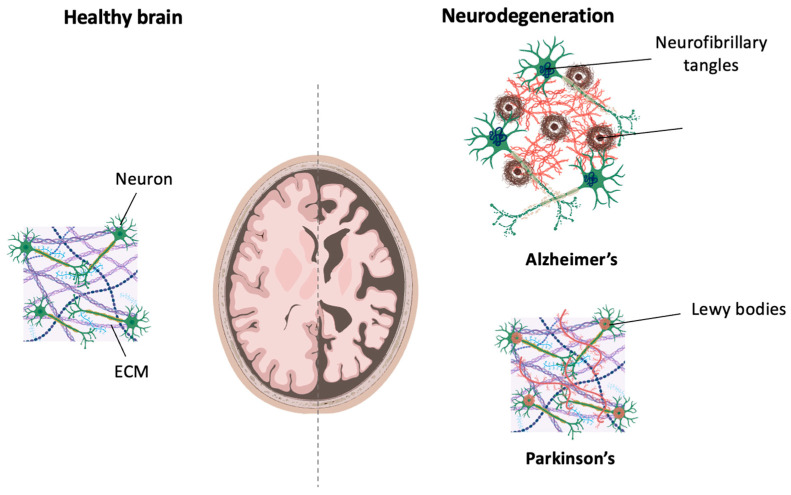
Schematic diagram showing a comparison between healthy brain tissue (**left**) and brain tissue affected by either Alzheimer’s or Parkinson’s disease (**right**). Alzheimer’s is characterized by the formation of extracellular Aβ plaques and intracellular neurofibrillary tangles, while the main hallmark of Parkinson’s is the intracellular accumulation of α-synuclein in the form of Lewy bodies. Created with BioRender.com.

**Figure 2 biomolecules-14-00069-f002:**
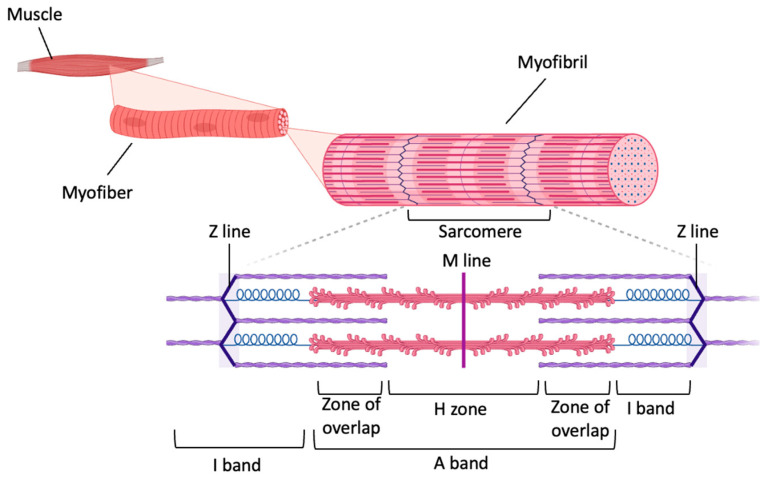
Schematic diagram showing the hierarchical organization of skeletal muscle tissue. Muscle is composed of multinucleated cells known as myofibers, which contain multiple myofibrils. The latter are comprised of a series of sarcomeres, the contractile unit of muscle tissue. The sarcomere is a highly organized structure formed of actin and myosin filaments, the main orchestrators of muscle contraction. Created with BioRender.com.

**Figure 3 biomolecules-14-00069-f003:**
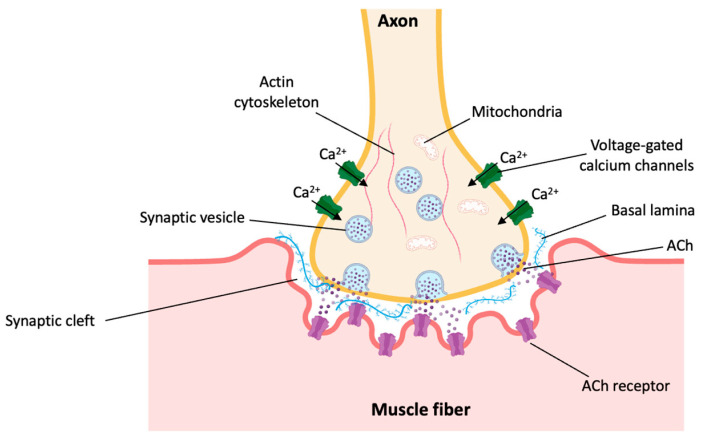
Schematic diagram of the neuromuscular junction (NMJ), which is formed of a motor nerve terminal, the synaptic cleft, and the innervated muscle fiber. The arrival of an action potential to the nerve terminal causes the opening of voltage-dependent calcium channels. The influx of calcium triggers the release of acetylcholine (ACh) from synaptic vesicles into the synaptic cleft. ACh diffuses through the basal lamina and binds acetylcholine receptors (AChRs) present in the post-synaptic element of the NMJ. The molecular events that are consequently triggered lead to the contraction of muscle tissue. Created with BioRender.com.

**Table 1 biomolecules-14-00069-t001:** Physio-chemical mechanisms used in the synthesis of inherently dynamic materials.

Material Re-Modeling Mechanism	Type of Interaction	Crosslinking Mechanism	Molecular Mechanism	Advantages	Examples	References
Reversible crosslinksReversible crosslinks	Non-covalent interactions	Host–guest complexes	Macrocyclic hosts with hydrophobic cavities and hydrophilic external surfaces (cyclodextrins, curcubit[*n*]urils, and calix[*n*]arenes) act as host molecules that encapsulate hydrophobic guest molecules, thus forming stable host–guest complexes	Specificity of the host–guest complexEase of reactionApplied to a diverse range of materials	Functionalization of hyaluronic acid (HA) with mono-acryloyl cyclodextrin and subsequent complexation with either adamantane or cholic acid via host–guest chemistry. The studies performed with these materials showed that crosslinks with a large dissociation rate constant facilitated cell spreading and mechanosensing.	[75,76,77,78]
Hydrophobic interactions	Driven by the repulsion between hydrophobic groups and the aqueous environment	Ease of preparationExcellent mechanical and self-healing properties	α-helical coiled-coil peptide hydrogelsAmphiphilic block copolymers	[79,80,81,82]
Hydrogen bonds	Secondary interactions that are weak in isolation but that lead to the formation of hydrogels with dynamic reversible crosslinks when present at significant numbers	Reliable and adaptable Self-healing Toughening effect through dissipation of external energy	Injectable four-arm PEG functionalized with either adenine or thymine. After mixing, a hydrogel was formed through hydrogen bonding between the nucleobases.Hydrogel formation via β-sheet assembly	[79,83,84]
Ionic interactions	Attractive or repulsive forces between charged molecules	Good solubilityRapid gelation	Alginate hydrogels crosslinked with divalent ions such as calcium	[79,84,85,86]
Covalent interactions	Dynamic covalent networks	Boronate esterReaction between a diol and boronic acid 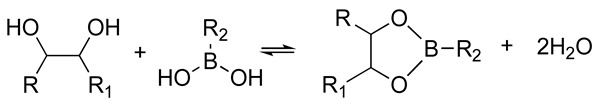 Diels-AlderReaction between an electron-rich diene, most commonly a furan, and an electron-poor dienophile, frequently a maleimide 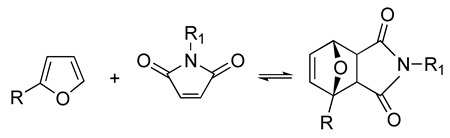 OximeReaction between an alkoxyamine and aldehyde or ketone, generating an oxime. Slower stress relaxation than other dynamic crosslinking reactions. 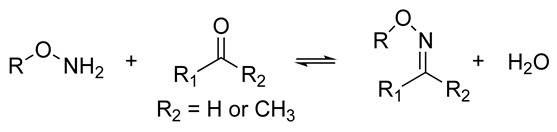 Thiol–disulfide exchangeSubstitution reaction where the thiol and disulfide groups are exchanged between different chemical species. The thiolate anion acts as a nucleophile. 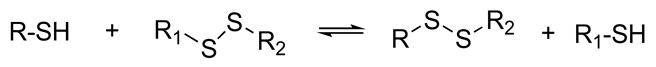 Imine crosslinkingReaction between an amine and an aldehyde or ketone, generating an imine 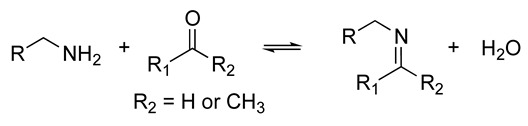 HydrazoneReaction between a hydrazine or hydrazide and aldehyde or ketone, generating a hydrazone linkage 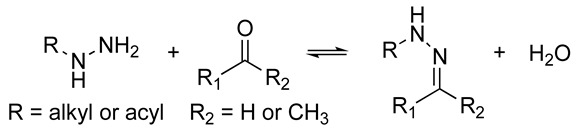	The resulting hydrogels can be tuned to have a similar viscoelasticity to that of the native human ECMDynamic covalent bonds can break and form on timescales that are comparable to those of cell-based matrix remodelingThese reactions can be carried out at physiological pH and temperatureThese reactions proceed at relatively fast speeds, and their kinetics can be tailored to produce hydrogels with pre-determined viscoelastic properties	Self-healing hydrogel formed of a copolymer of 2-acrylamidophenylboronic acid (2-APBA) and *N,N*-dimethylacrylamide (DMA) mixed with poly (vinyl alcohol) (PVA) Self-healing dextran hydrogels formed via reaction between fulvene-modified hydrophilic dextran (diene) and dichloromaleic-acid-modified poly(ethylene glycol) (PEG) Sodium alginate hydrogel with tunable stress relaxation via reaction between alkoxyamine-functionalized alginate and aldehyde-containing oxidized alginate. The resulting hydrogel had calcium-mediated and oxime crosslinking, which led to a greater degree of tunability. Dynamic covalently crosslinked keratin hydrogels formed via thiol–disulfide exchange. The hydrogels showed injectability, self-healing, and redox-responsive capacity. Collagen hydrogels crosslinked with imine bonds had greater stress relaxation rates than collagen crosslinked with methacrylate bonds. The faster stress relaxation promoted cell spreading within one day. Formation of hydrazone bond between an aliphatic aldehyde-terminated multi-arm PEG and an aliphatic hydrazine-terminated multi-arm PEG, resulting in highly viscoelastic gels that promoted 3D cell spreading and the formation of multinucleate structures with a myotube-like morphology	[74,79,87,88,89,90,91,92,93]
Chemically responsive moieties		Hydrolyzable ester linkages	Ester bonds are spontaneously hydrolyzed in water 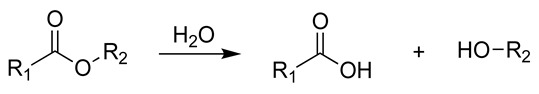	Not dependent on the levels of enzyme present in the sample	Hyaluronic acid crosslinked with PEGDA to make the hydrogel susceptible to hydrolysisPolyesters, polyethers, polycarbonates	[94,95,96,97]
	Enzymatically-degradable peptide crosslinks	Degradation of hydrogels by cell-secreted enzymes 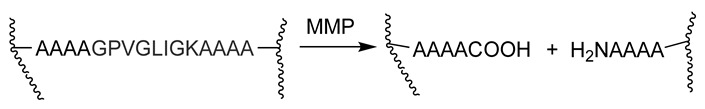	Restricts degradation to regions of cell invasion Cell-mediated mechanism can couple the degradation rate with the rate of tissue formationThe rate of degradation can be tuned by altering the peptide sequence	MMP-degradable hydrogels formed via reaction between 4arm-PEG tetravinyl sulfone and bis-cysteine peptide with an MMP-sensitive sequence	[98,99]

**Table 2 biomolecules-14-00069-t002:** Dynamic biomaterials tunable on demand upon application of external stimuli.

Stimuli	Molecular Mechanism	Principle	Advantages	Disadvantages	References
Temperature	Thermoresponsive polymers with upper or lower critical solution temperature (UCST or LCST), below which they are either insoluble or soluble, respectively	At the critical temperature, a change in the polymer solubility occurs and causes a change in molecular conformation	Easy to control the culture temperature	Temperature changes can affect cell viability and metabolic processes	[101,102]
Light	Incorporation of a photosensitive molecule such as azobenzene Photocleavable crosslinkers	Illumination changes the molecular conformation or induces a chemical reaction in the photoactivated moiety	Contact-free, easy, and precise on-demand control of stimulation	Possible chromophore toxicity if not covalently bound to the polymer Potential phototoxicity	[101]
Ultrasound	Crosslinker cleavage that causes changes in stiffness Disassembly of vesicles that leads to cargo release Gel transitions	High-frequency waves cause a rise in temperature and cavitation effects (growth and shrinkage or implosion of micro bubbles). The resulting pressure causes an alteration in the mechanical properties of the material.	Ease of application using an ultrasound transducer May be able to use existing material design without incorporating additives	Limited range of parameters that can be tuned in response to ultrasound	[101,103,104]
Electric field	Conductive polymers or incorporation of conductive materials into the polymer The deformation of a material in an electric field is influenced by variations in osmotic pressure, pH, electrode position, and the applied voltage	The application of an external electric field causes changes in the structural and mechanical properties of the material	Allows the generation of soft robotic materials that closely mimic human motor function Wide variety of electroresponsive materials	Difficulties optimizing the magnitude of the electric current Some of these materials have a low biocompatibility Many of the materials have poor mechanical strength and are brittle Cells respond to the electric field	[101,103,105,106,107]
Magnetic field	Materials that contain ferromagnetic structures	The application of a magnetic field triggers a reorganization in the magnetic structures that leads to changes in polymer structure, viscosity, or stiffness	Fast and reversible material modulation Magnets are easily included in culture systems	Possible leakage and toxicity of magnetic particles Materials containing magnetic particles are often opaque, making observing real-time changes in cell behavior challenging	[101,103,108,109,110]
Strain	Fiber reorganization and alignment, non-covalent interactions between fibers	The application of a mechanical strain leads to changes in the mechanical properties of the material	The strain can be externally applied or generated by cells Wide variety of materials with these properties	The application of strain can cause alterations in the topography of the material, such as the generation of wrinkles, which may serve as confounding variables It is difficult to separate the effects of cell-generated vs. user-generated strain effects	[101,103,111,112,113,114,115,116,117]

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
