# Peer review of "Leveraging Biomaterial Platforms to Study Aging-Related Neural and Muscular Degeneration"

_biomolecules, 2024, doi:10.3390/biom14010069_

Round 1
Reviewer 1 Report
Comments and Suggestions for Authors
In general, the manuscript describes the development made in the creation of 3D models of aging-related neuronal and muscular degeneration and it provides a platform on the future directions for the researchers who are working in this area. as overall, the review article is organized well, discussions are adequate. Hence it is recommended for publication in the journal of 'Biomolecules' in the present form.
Comments on the Quality of English LanguageGood
Author Response
Thank you very much for your positive feedback, it is very much appreciated.
Reviewer 2 Report
Comments and Suggestions for Authors
A manuscript of the article “Leveraging Biomaterial Platforms to Study Aging-Related Neural and Muscular Degeneration”, prepared by two authors, was submitted for review by the reviewer.
The text discusses current issues of creating three-dimensional models of biological tissues that can reliably simulate the effects of age-related neuronal and muscle degeneration.
First of all, I would like to point out the very good quality of the text, from the point of view of a physicist reader working in the field of 3D printing for tissue engineering. Working on similar problems, I realize that my knowledge was not deep enough - I learned a number of important aspects in the development of tissue models of neuronal tissue, especially using customizable dynamic materials.
I have a couple of recommendations for the manuscript.
In my opinion, Section 2 lacks a broader listing of naturally occurring materials that are used to form model and scaffold structures for nerve cells. In my opinion, in the text of the manuscript it is worth mentioning in more detail about materials based on chitin and its derivatives (chitosan), which are often used to form structures for neuronal tissue. Also, I think it is important to note in the publication the possibility of using hydrogels with a combined composition, for example, chitosan with polylactide copolymers, which make it possible to form strong hydrogel structures with the necessary mechanical and strength parameters. If these materials, in your opinion, are not suitable for fulfilling their role, then I suggest mentioning them and justifying why they are not suitable.
In Section 4, in the part of the text where you list methods for forming microstructures for micromolding, I recommend specifically mentioning laser technologies as an affordable alternative to the lithographic approach for the formation of micropatterns. In my opinion, laser microstructuring or laser 3D printing (two-photon polymerization) are sometimes available effective methods for creating microstructures with a unique unit design.
Overall, in my opinion, the submitted manuscript should undoubtedly be accepted for publication in the journal, and the information presented in the article will undoubtedly be in demand by the scientific community.
Author Response
Thank you very much for your positive feedback and comments, we are very happy to hear that the information included in this review was so useful for you. Thank you very much for your advice, you make very good points about the addition of information about chitosan and other biopolymers, as well as the laser-based microfabrication. We have added information and examples about the use of chitosan and alginate in neuronal bioengineering (section 3). We have also added information about micropatterning via laser ablation, as well as advantages and pitfalls (section 4).
Reviewer 3 Report
Comments and Suggestions for Authors
In the manuscript titled ¨Leveraging Biomaterial Platforms to Study Aging-Related Neural and Muscular Degeneration¨, the authors summarize the crucial implication of 3D cultures that recapitulate aging with a special focus on neural and muscular cells. This review is relevant because of the urgency to develop novel human preclinical in vitro models. The manuscript is well written.
Author Response

(The authors gave the same response as above.)
